

# Allometric shell growth in infaunal burrowing bivalves: examples of the archiheterodonts *Claibornicardia paleopatagonica* (Ihering, 1903) and *Crassatella kokeni* Ihering, 1899

Damián Eduardo Perez[*] and María Belén Santelli[*]

División Paleoinvertebrados, Museo Argentino de Ciencias Naturales "Bernardino Rivadavia", Buenos Aires, Argentina

[*] These authors contributed equally to this work.

## ABSTRACT

We present two cases of study of ontogenetic allometry in outlines of bivalves using longitudinal data, a rarity among fossils, based on the preserved post-larval record of shells. The examples are two infaunal burrowing bivalves of the southern South America, *Claibornicardia paleopatagonica* (Archiheterodonta: Carditidae) (early Paleocene) and *Crassatella kokeni* (Archiheterodonta: Crassatellidae) (late Oligocene–late Miocene). Outline analyses were conducted using a geometric morphometric approach (Elliptic Fourier Analysis), obtaining successive outlines from shells' growth lines, which were used to reconstruct ontogenetic trajectories. In both taxa, ontogenetic changes are characterized by the presence of positive allometry in the extension of posterior end, resulting in elongated adult shells. This particular allometric growth is known in others infaunal burrowing bivalves (*Claibornicardia alticostata* and some *Spissatella* species) and the resulting adult morphology is present in representatives of several groups (e.g., Carditidae, Crassatellidae, Veneridae, Trigoniidae). Taxonomic, ecological and evolutionary implications of this allometric growth pattern are discussed.

## INTRODUCTION

According to the Gould-Mosimann school (defined by *Klingenberg, 1998*), 'allometry' is the association between size and shape. The concept of allometry implies variation of a trait associated with variation of the overall size of an organism (*Klingenberg, 1998*). Size of an organism can be determined by its own biological growth (or ontogeny), and in these cases, allometry is the covariation between shape and growth through its life-span. This allometry is known as "ontogenetic allometry" (*Klingenberg, 1996a*; *Klingenberg, 1998*). Studies on ontogenetic allometry mainly use "cross-sectional" data (each individual is measured at a single stage, and an average allometric trajectory is estimated from a composite sample from many individuals). Some ones use "longitudinal" data

Corresponding author
Damián Eduardo Perez,
trophon@gmail.com

(e.g., *Klingenberg, 1996b*; *Maunz & German, 1997*) (each individual is measured multiple times during their growths, and individual variability in allometric trajectories is obtained). Cases of "cross-sectional" data (sensu *Klingenberg, 1996b*) are frequent in paleontological studies, for example in trilobites (see *Hughes, Minelli & Fusco, 2006* and references herein), Cambrian arthropods (e.g., *Haug et al., 2011*), crinoids (e.g., *Brower, 1988*), gastropods (e.g., *Gould, 1966a*), diapsids (e.g., *Ezcurra & Butler, 2015*), dinosaurs (e.g., *Horner & Goodwin, 2006*; *Horner & Goodwin, 2009*), or mammals (e.g., *Christiansen, 2012*). "Longitudinal" studies (sensu *Klingenberg, 1996b*) are not possible for many fossil organisms, but are viable in organisms with accretionary growth. Some examples are shelled molluscs (*Urdy et al., 2010*), brachiopods (*Rudwick, 1968*; *Ackerly, 1989*; *Tomašových, Sandra & Labarbera, 2008*), or ammonoids (*Korn, 2012*; *Korn, 2017*; *De Baets, Klug & Monnet, 2013*). Some researches often remain focused on adult stages, not taking into account the complete ontogeny, what is necessary for a more holistic view.

Bivalves show accretionary growth in their shells where the mantle adds constantly new layers of calcium carbonate to the edge (*Pannella & Maclintock, 1968*). Therefore, they preserve in their shells a complete record of external traits of their post-larval life-spans (*Crampton & Maxwell, 2000*), making them a source of "longitudinal" data (sensu *Klingenberg, 1996b*) for construction of ontogenetic trajectories. In a pioneer contribution, *Crampton & Maxwell (2000)* elaborated a methodology to explore this particular growth in bivalves. They re-constructed the ontogenetic trajectories of New Zealand species of *Spissatella* (Bivalvia: Crassatellidae) and related their allometric growth to macroevolutionary trends in the clade.

From the paleoecological point of view, fossil bivalves are one of the most valuable tools, as different morphologies of bivalve shell are strongly related to modes of life and environmental characteristics (*Stanley, 1970*). Infaunal burrowing habit of life is the most extended among bivalves, consisting of the penetration of soft substrates by means of a pedal locomotion while maintaining a life position of, at least, partial burial (*Stanley, 1970*).

Geometric morphometrics is a very useful tool for study of allometry and ontogeny (*Zelditch, Bookstein & Lundrigan, 1992*; *Fink & Zelditch, 1995*; *Mitteroecker et al., 2004*; *Mitteroecker, Gunz & Bookstein, 2005*; *Monteiro et al., 2005*; among others, see a revision on this topic in *Adams, Rohlf & Slice, 2013*). Morphometric methods are objective, reliable and repeatable tools for quantify patterns of shape changes (*Brown & Vavrek, 2015*). Geometric morphometric allows visually strong graphical representations of allometry studies (*Adams, Rohlf & Slice, 2013*). In particular, outline shape analyses allow to study the variation in this key character, the outline, which reflects autoecological features in bivalves according to *Stanley (1970)* and *Stanley (1975)*. The aim of this contribution is to study ontogenetic series in two examples of infaunal burrowing bivalves, *Claibornicardia paleopatagonica* (*Ihering, 1903*) (Archiheterodonta: Carditidae) and *Crassatella kokeni* (*Ihering, 1899*) (Archiheterodonta: Crassatellidae). Variability in shape of these two bivalves led previous authors to define new species based on possible juvenile specimens, *Venericardia camachoi* (Vigilante, 1977) and *C. patagonicus* (*Ihering, 1907*) (nowadays considered as synonymies of *C. paleopatagonica* and *C. kokeni*, respectively). Presence of allometric growth is tested and changes in shape in these species, and changes present in other infaunal bivalves,

as well as their paleoecological implications, are discussed. Also, this contribution is an attempt to apply and to expand the methodology developed by *Crampton & Maxwell (2000)*. As is already mentioned by *Crampton & Maxwell (2000)*, *Gould* (*1989*, p. 537) noted that "Natural history is a science of relative frequencies"; and as these authors indicated "advance in many fields of palaeontological debate requires compilation of detailed observations across diverse fossil groups and time spans" (*Crampton & Maxwell, 2000*, p. 400). The present is a contribution for thickening the literature of cases studying allometry patterns, and this is necessary since a debate addressing the relative frequencies of different phenomena only advance through the compilation of such cases.

## MATERIALS & METHODS

### Terminology and theoretical background

All terms regarding allometry follow the definitions provided by *Klingenberg (1998)*. Positive allometry refers to a trait that increases respect to another one (a positive deviation to expected isometry), and negative allometry is the opposite. Geometric Morphometrics and Elliptic Fourier Analysis (EFA) terminologies are explained in *Kuhl & Giardina (1982)*, *Lestrel (1997)*, and *Crampton (1995)*.

According to *Crampton & Maxwell (2000)*, two outlines with identical shapes and differing only in size will occupy the same point in a morphospace as the distance in this space is a measure of shape difference, a statement that was followed to perform the analysis in this paper.

Bivalve species studied herein are considered as infaunal free burrowing bivalves because they live under the water/sediment interphase and they are not-attached by their byssus. This categorization was described by *Stanley (1970)* and its followed in this contribution. From this point onwards, this mode of life will be called as "infaunal".

### Taxon sampling

Allometric growth was studied in two species from the Cenozoic of Argentina, *C. paleopatagonica* (*Ihering, 1903*) (Archiheterodonta: Carditidae) (Fig. 1A) and *C. kokeni* (*Ihering, 1899*) (Archiheterodonta: Crassatellidae) (Fig. 1B). Archiheterodonts are non-siphonate bivalves, being mainly restricted to shallow infaunal free burrowing. All fossil shells used in this study are housed at Museo Argentino de Ciencias Naturales "Bernardino Rivadavia" (MACN-Pi and CIRGEO-PI) and Paleontological Collection of Universidad de Buenos Aires (CPBA). Sampling details are summarized in Data S1.

The carditid species represents the most ancient record for its genus, being recorded in the early Danian of Patagonia (Argentina), in the Roca, Jagüel and Salamanca formations (Río Negro, Neuquén and Chubut provinces) and was recently included by *Pérez & Del Río (2017)* in the genus *Claibornicardia* (*Stenzel & Krause, 1957*). This taxon is also recognised in the late Paleocene–early Oligocene of North America and Europe. In these analyses 15 well-preserved shells of *C. paleopatagonica* from Puesto Ramírez (Salamanca Formation, Río Negro Province) (MACN-Pi 5197) were used. The specimen previously assigned to *Venericardia camachoi* by Vigilante (1977) is also included in MACN-Pi 5197.

 

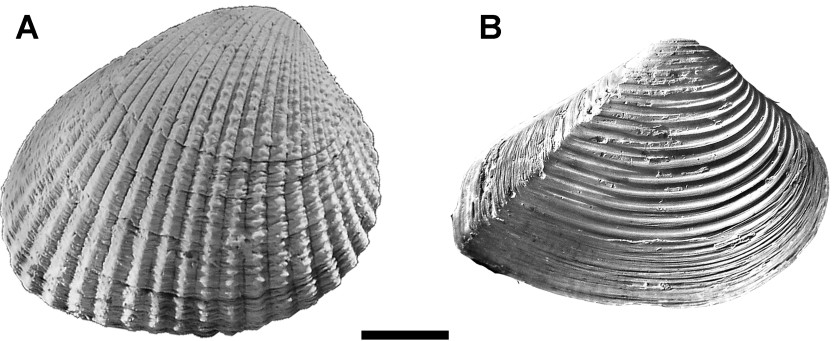

**Figure 1** **Case-studies of this work.** (A) MACN-Pi 5197, *C. paleopatagonica* (*Ihering, 1903*) (Puesto Ramírez, Salamanca Formation, Early Danian) in lateral view. (B) MACN-Pi 3576, *C. kokeni Ihering, 1899* (mouth of Santa Cruz River, Monte León Formation, Early Miocene) in lateral view. Scale bar = 10 mm. Photo credit: the authors.

*C. kokeni* is the most abundant crassatellid from the Cenozoic of Patagonia (Argentina), being represented in the San Julián, Monte León, Camarones and Puerto Madryn formations (late Oligocene–late Miocene, Chubut and Santa Cruz provinces). The systematics of this species was reviewed by *Santelli & Del Río (2014)*, who regarded *Crassatellites patagonicus* (*Ihering, 1907*) as a junior synonymous of *C. kokeni*. For our analyses, 32 well-preserved shells of *C. kokeni* were used (including those previously assigned to *C. patagonicus*). These specimens come from Cañadón de los Artilleros, Punta Casamayor, Cabo Tres Puntas (late Oligocene–early Miocene, San Julián Formation, Santa Cruz Province); mouth of Santa Cruz River, Estancia Los Manantiales, Cañadón de los Misioneros, Monte Entrada (early Miocene, Monte León Formation, Santa Cruz Province); Camarones (early Miocene, Camarones Formation, Chubut Province), and Lote 39 (late Miocene, Puerto Madryn Formation, Chubut Province) (MACN-Pi 325–327, 331–332, 3576, 3600, 3907, 4775, 5374–5376; CIRGEO-PI 1501–1502; and CPBA 9404).

## Elliptic Fourier analysis

The Elliptic Fourier Analysis (*Kuhl & Giardina, 1982*) method was chosen to analyse the outlines of our examples because it allows to work with the variation presents in valves shape. The methodology employed to obtain different outlines is derived from *Crampton & Maxwell (2000)* criteria. Each valve was digitally photographed in an inclined position with their growth lines placed parallel to the surface (Fig. 2A). The outlines obtained in different angles, regarding to the surface, were limited by coarse growth lines across the entire shell (Fig. 2B). Strict chronological ages of each individual have not been established, but previous analyses have well found a strong correlation between ages (based on the use of stable isotopes) and growth lines (*Jones, 1988*; *Brey & Mackensen, 1997*; *Jones & Gould, 1999*; *Lomovasky et al., 2002*). As a result, growth lines are a good proxy for the chronological age of specimens, and size is an estimation for relative time. In *C. paleopatagonica* annual growth lines are noticeable but in *C. kokeni* they are not so evident, being perceptible only in part of specimens' shells. For this species, outlines were taken at intervals of 10 mm

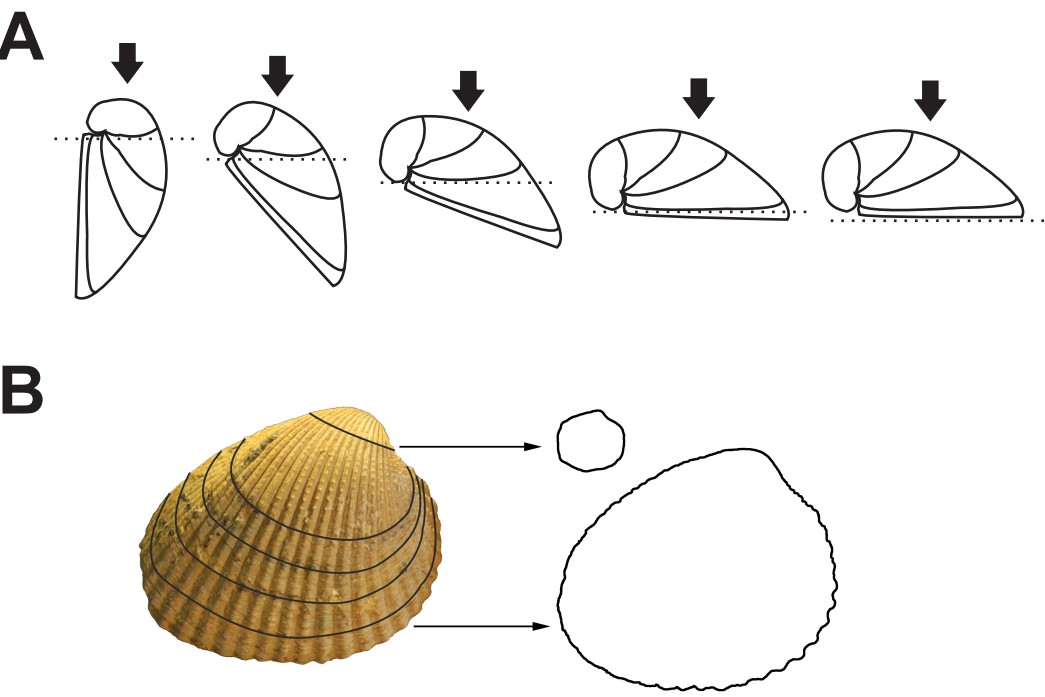

**Figure 2** **Applied methodology to obtain successive outlines of a single valve.** (A) One valve oriented in different angles. Dotted lines indicate the parallel position of valves regarding to surface, arrows indicate position of digital camera. (B) Example of successive outlines captured in one specimen (MACN-Pi 5197). First and last outline illustrated. Photo credit: the authors.

along the axial length, following the procedure undertaken by *Crampton & Maxwell (2000)* for *Spissatella*. This methodology allows to design an age-structured analysis for our data.

From a digitization procedure using a digital camera, 62 outlines were obtained from *C. paleopatagonica*, and 74 outlines from *C. kokeni*. Noise generated by external sculpture was removed from outlines with an image-edition software (Adobe Photoshop CS5) (following *Crampton, 1995*). Right valves were mirrored on the horizontal axis taking advantage of the equivalve character of shells, and the analysis was performed only with left valves. The outlines were grouped into three growth categories: "less than two", "two to four", and "more than four", each one indicating the number of precedent coarse growth lines. In the case of *C. kokeni*, due to different geographic and stratigraphic provenance of the studied specimens, four geological categories were established to group outlines: 'Monte León', 'Camarones', 'Puerto Madryn', and 'San Julián', each one representing the geological provenance of the material.

For each individual, chain codes were registered along the contour to calculate the Elliptic Fourier Descriptors (EFDs). Total Fourier power was calculated to estimate the optimal number of harmonics required for the analysis. The Fourier power of a harmonic is proportional to its amplitude and provides a measure of the amount of shape described by that harmonic (*Crampton, 1995*). A series of harmonics can be truncated when the value

**Table 1  Univariate regression analysis between size (area) and shape (principal component).**

|  | Slope | Intercept | *p*-value |
|---|---|---|---|
| *C. paleopatagonica* (PC1) | 763.09 | −387001.83 | 1.248E−15 |
| *C. kokeni* (PC2) | 879.3 | −208871.5 | 8.149E−08 |

of average cumulative Fourier power reaches the 99% of the average total power (sum of the total harmonics used). The optimal number for this case was stablished in ten harmonics for *C. paleopatagonica*, and seven harmonics for *C. kokeni*. Outlines were normalized to discard effects of rotation, translation and size, using the parameters of the ellipse defined by the first harmonic (First Harmonic Ellipse method). Therefore, three of the four EFDs describing the first harmonic ellipse are constant for all the outlines (*Crampton, 1995*). The software Shape 1.3v (*Iwata & Ukai, 2002*) was used for all the analysis.

## Morphospace construction and regression analysis

A Principal Component Analysis (PCA) was performed from the variance–covariance matrix of normalized coefficients (Data S2 and Data S3 shows normalized Fourier coefficients for each outline and for each taxon, respectively). The shapes of the shell for mean and extreme morphologies (the latter are representations of specimens with score values corresponding to −two and +two standard deviations from the centre for each component) were reconstructed from the normalized coefficient mean values of the EFDs using the inverse Fourier transformations (*Iwata & Ukai, 2002*) and plotted alongside the morphospace reconstruction. The growth and geological categories previously defined were both plotted on the PCA. Also, a Univariate Regression Analysis (URA) between sizes (obtained from the two-dimensional area of each outlines) and shapes using the principal components in both study-cases was conducted. The components were selected exploring the morphological variance obtained from PCA. The morphospace construction were performed using PAST 3.19 (*Hammer, Harper & Ryan, 2001*), and the URA using R environment (*R Core Development Team, 2017*).

## RESULTS

### *C. paleopatagonica* allometric growth

The first three components of PCA explain 74.02% of the total variance (Fig. 3A). The first component (PC1) explains 46.55% of variance and represents the transition between subcuadrate (negative extreme) to subrectangular/subelliptic (positive extreme) outlines, with a posterior-ventral expansion. The second component (PC2; 20.16% of variance) accounts for changes in convexity and width of umbones (more rounded umbos towards positive values and less rounded towards negative values). The third component (PC3; 7.3 % of variance) captures variation in concavity of the lunular area (more concave lunule towards negative values and more convex lunule towards positive values). The URA between size and PC1 (selected because this component shows a transition between subcuadrate and subrectangular outlines) is significant (*p*-value<0.001) (Fig. 4A). Results of PCA and URA analyses are included in Table 1 and Data S4.

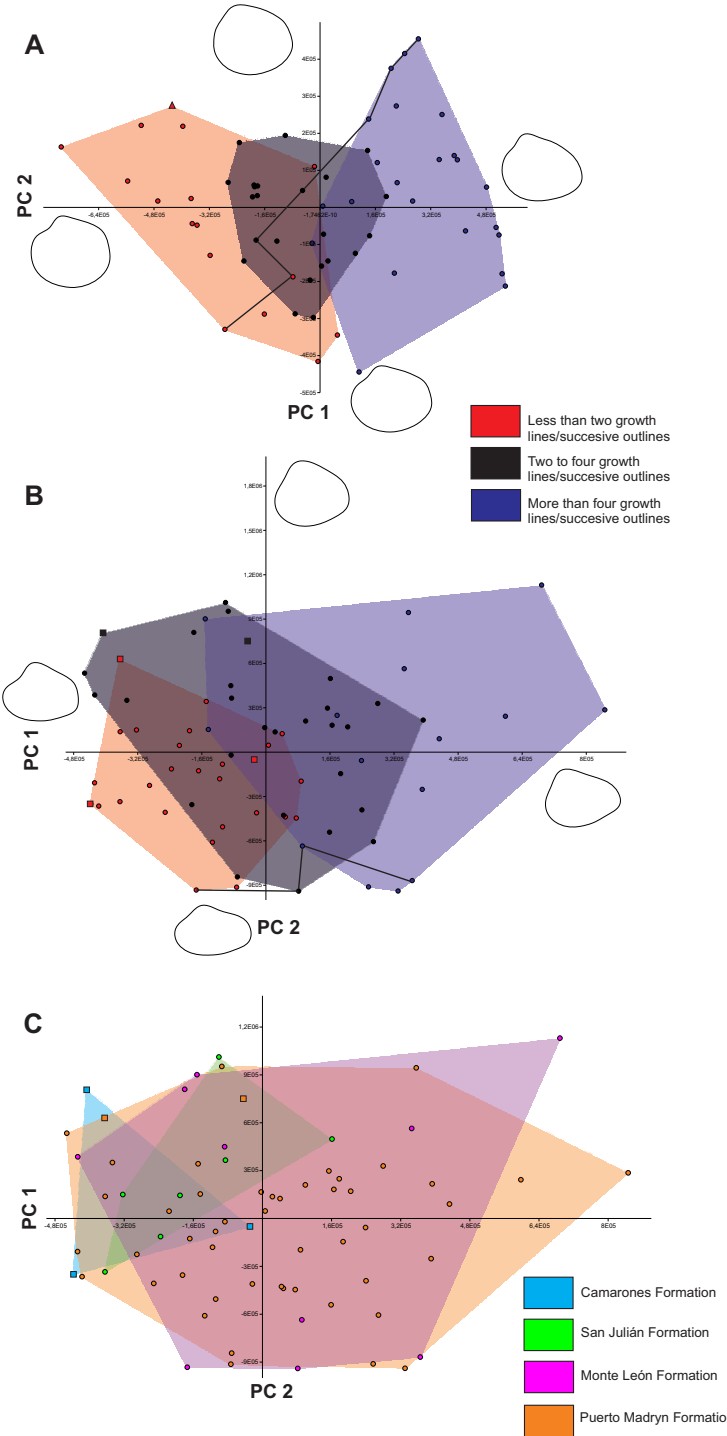

**Figure 3  Results of principal component analyses.** (A) *C. paleopatagonica* arranged by ontogenetic stage. (B) *C. kokeni* arranged by ontogenetic stage. (C) *C. kokeni* arranged by stratigraphic procedence. Color legends and the extreme morphologies of each principal component are illustrated in the graph. Black lines in A and B show ontogenetic trajectories of a selected specimen. Triangles indicate specimen previously assigned to *Venericardia camachoi* and squares indicate specimen previously assigned to *C. patagonicus*.

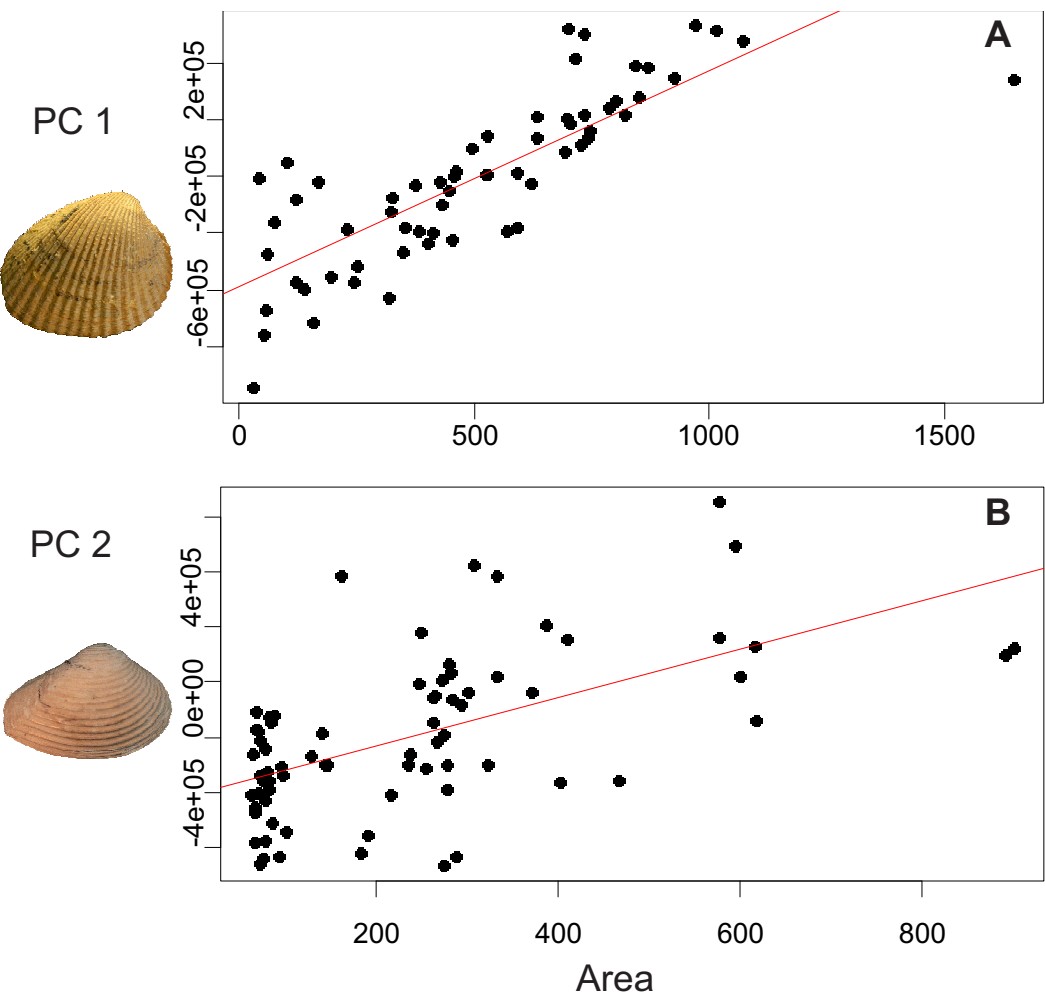

**Figure 4** **Results of univariate regression analyses, between area (size) and principal components (shapes).** (A) includes first principal component obtained from *C. paleopatagonica*, and (B) includes second principal component obtained from *C. kokeni*. Red line indicates trend line. Photo credit: the authors.

Growth categories plotted in the obtained morphospace show a transition across PC1 from juvenile to adult outlines. Variation across life-span in *C. paleopatagonica* can be distinguished in the successive outlines of each individual. Juvenile outlines are strongly rounded and shows subcentrally placed umbones. Towards more aged shells, an increase in the projection of posterior end is recognisable. Adult shells of this species have subrectangular to subelliptic outlines with anteriorly placed umbones. A reconstructed ontogenetic trajectory can be observed in Fig. 3A linking different stages of the same specimen in the morphospace (this ontogenetic trajectory was obtained from a single actual specimen, from which the largest number of outlines were acquired). Different allometric variation can be detected when overlapping extreme outlines of PC1. Posterior end has positive allometry, while the dorsal and anterior-ventral margins have negative allometry (Fig. 5A).

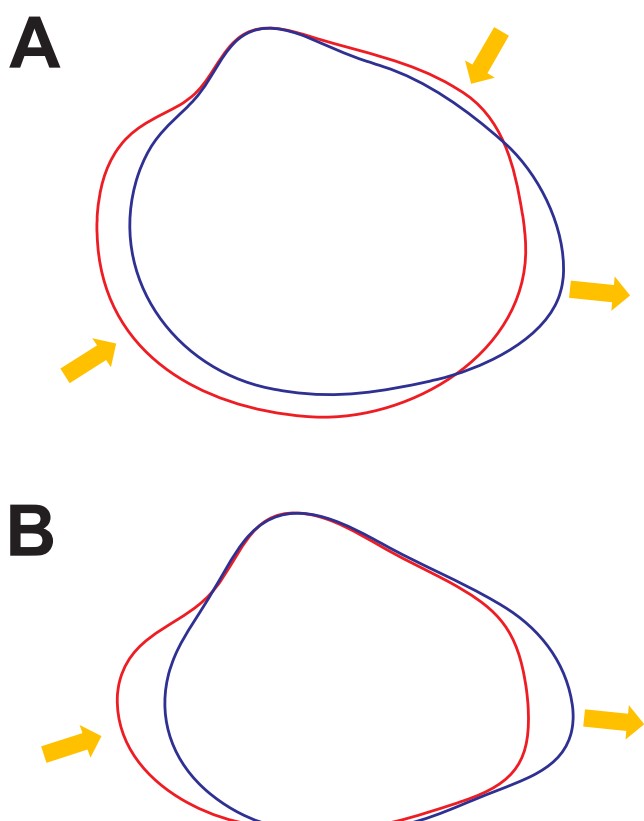

**Figure 5  Overlapping of extreme outline configurations.** (A) *C. paleopatagonica*. (B) *C. kokeni*. Red outline, juvenile specimens. Blue outline, adult specimens. Arrows indicate positive or negative allometry.

### *C. kokeni* allometric growth

In this case, the first three components of PCA explain 90.72% of the total variance (Fig. 3B). The first component (PC1; 66.66% of variance) shows variation between outlines with subcentrally placed umbones and outlines with anteriorly placed umbones. The second component (PC2; 19.27% of variance) reflects variation between more subtriangular and more subrectangular outlines. The third component (PC3; 4.79% of variance) is associated to variation between less and more truncated posterior end of valves. In this case, the PC2 was selected to the URA test, because this component explains the transition between less and more elongated outlines. The URA shows more scattered points on the graphs than *C. paleopatagonica*, which could be related to the different geological provenance of shells. Nevertheless, the result is significant (*p*-value<0.001) (Fig. 4B). Results of PCA and URA analyses are included in Table 1 and Data S5.

Geological categories show a non-structured arrangement when they are plotted in the morphospace. The best sampled categories ('Monte León' and 'Puerto Madryn') occupy virtually the whole morphospace (Fig. 3C). Growth categories reflect a transition across the PC1 from juvenile to adult outlines. Juvenile outlines of *C. kokeni* are strongly subtriangular with pointed umbones, whereas adult outlines are markedly subrectangular having more

rounded umbones. The reconstructed ontogenetic trajectory (Fig. 3B), obtained in the same way as the previous case, and the overlapping of both extreme outlines of PC2 shows an allometric variation similar to those observable in *C. paleopatagonica* (Fig. 5B). Specimens originally assigned to *C. patagonicus* by previous authors fall into the juvenile sector of the morphospace (Fig. 3B).

## DISCUSSION

### Morphological change across life-span in *C. paleopatagonica* and *C. kokeni* and related species

An allometric growth pattern shared by both species, *C. paleopatagonica* and *C. kokeni* was found in the analyses. Both taxa have positive allometry detected in the extension of posterior end, resulting in elongated adult shells. The study of ontogeny in bivalves had evidenced that some species show allometric growth in certain characters (*Stanley, 1975*; *Stanley, 1977*; *Tashiro & Matsuda, 1988*; *Savazzi & Yao, 1992*) and the morphological change recorded herein is also documented in other phylogenetically related infaunal bivalves. Subquadrate juvenile and elongated adult specimens of the carditid *Claibornicardia alticostata* (*Conrad, 1833*) have a similar allometric variation (*Stenzel & Krause, 1957*, and D Perez, pers. obs., 2015 on syntypes ANSP 30562). *Crampton & Maxwell (2000)* described a similar variation in some representatives of the crassatellid genus *Spissatella*, especially in the species *S. subobesa* (*Marshall & Murdoch, 1919*) and *S. poroleda* (*Finlay, 1926*).

### Elongated adult morphology in other infaunal bivalves

Ontogenetic trajectories have not been described in other infaunal bivalves. However, the same elongated adult morphology described here is known. Among archiheterodonts, the morphology documented for adult shells of *C. paleopatagonica* and *C. kokeni* can be observed in species of the genera *Megacardita Sacco, 1899* (*La Perna, Mandic & Harzhauser, 2017*); *Neovenericor Rossi de García, Levy & Franchi, 1980* (*Pérez, Alvarez & Santelli, 2017*); *Venericor Stewart, 1930* (*Gardner & Bowles, 1939*); and *Bathytormus Stewart, 1930* (*Wingard, 1993*; *Santelli & Del Río, 2014*). Among other bivalve groups, this adult morphology is also recorded in species of the Veneroidea and Palaeoheterodonta. Some species of Veneridae genera as *Anomalocardia Schumacher, 1817*, *Lirophora Conrad, 1863*, *Chionopsis Olsson, 1932*, *Lamelliconcha Dall, 1902*, *Macrocallista Meek, 1876*, and *Antigona Schumacher, 1817*, among others, have adult shells with a projected posterior end and elongated outlines. Some Trigoniidae taxa lead this morphology to the extremes, with the development of wide and very projected posterior ends (e.g., *Francis & Hallam, 2003*). As an example, *Echevarría (2014)* found a strong allometric growth developing in two phases in the trigoniid *Myophorella garatei* (*Leanza, 1981*) with a strong extension of the posterior margin.

### Taxonomic implications of allometric growth

Differences between young and adult morphologies could have been be interpreted as taxonomic differences between species. In both studied cases, new species were proposed for specimens with young morphologies: *Venericardia camachoi* Vigilante, 1977 and

*C. patagonicus* (*Ihering, 1907*). These taxa fall into the variation representing young specimens of *C. paleopatagonica* and *C. kokeni*, respectively. The case of *C. kokeni* and *C. patagonicus* was already mentioned by *Santelli & Del Río (2014)*, being corroborated the synonymy in this study. Other examples are the carditids *Neovenericor paranensis* (*Borchert, 1901*) (late Miocene, Argentina), the adult morphology of which was described as *Venericor crassicosta Borchert, 1901* (*Pérez, Alvarez & Santelli, 2017*) and *Neovenericor ponderosa* (*Suter, 1913*) (late Oligocene, New Zealand), the young morphology of which was named *Venericardia caelebs* Marwick, 1929 (*Beu & Maxwell, 1990*). These results reflect that this allometric change (included into the instraspecific variation) must be considered in taxonomic revisions of similar infaunal bivalves. These examples show that a different outline is frequently considered an important feature for taxonomic recognition but ontogenetic variation is not always taken into account (*Alvarez & Pérez, 2016*).

## Ecological implications of the elongated adult morphology

According to *Stanley*'s experiments (*1970*), bivalve shells with streamlined outlines (cylindrical, blade-like, or disc-like) are the fastest burrowers. Elongated outlines could be related to fast burrowing in soft substrates but not in all cases. Also, *Stanley (1970)* established that moderately elongated burrowing species commonly use a large angle of rotation, having a strong forward component in their burrowing movement because of their eccentric axis of rotation. Elongated bivalves generally have a mode of life with the long axis in vertical position–for example, this is observed in living species of *Anomalocardia*–. Posterior portion of shell is directed to sediment surface, being achieved the elongated morphotype with a minimum of increase in shell growth, displacing the centre of gravity and the visceral mass of organisms to a deeper position (*Stanley, 1970*; *Crampton & Maxwell, 2000*). Other possibly related effects could be increasing in stability against scour (*Stanley, 1977*; *Stanley & Yang, 1987*; *Francis & Hallam, 2003*) or reduction of exposure and predation (*Crampton & Maxwell, 2000*; *Francis & Hallam, 2003*). One possible way to reach this morphology could be to exploit the positive allometry of posterior end through the ontogeny.

*Crampton & Maxwell (2000)* suggested that ontogenetic variation in *Spissatella* is an adaptation for life in more energetic environments with coarser substrates but these parameters were not explored in our data. Nevertheless, these conditions (along with others such as predation) may have played a part as selective pressures in the evolutionary history of these infaunal bivalves. Further stratigraphic structured analyses, including taphonomic and sedimentologic data, are needed to study these hypotheses.

## Evolutionary implications of allometric growth

Ontogenetic changes in the mentioned infaunal bivalves seem to be similar and perhaps, could be induced by similar conditions. Allometry plays a significant role in evolutionary trends of most lineages (*Gould, 1966b*; *Gould, 1977*; *Klingenberg, 1998*). The study of allometric changes is sometimes necessary for recognition of some cases of heterochronic processes (e.g., *Shea, 1983*; *McKinney, 1984*; *Mitteroecker, Gunz & Bookstein, 2005*). Heterochrony is the change in relative time of appearance of characters already presents

in ancestors (*Gould, 1977*; *McNamara, 1986*). Learning more about the ontogenetic trajectories and allometric changes present in different taxa is essential as the first step for heterochrony studies. These analyses require ontogenetic trajectories explored and phylogenetic relationships defined among species, being the cases like the ones described here fundamental and very important as a starting point.

## CONCLUSIONS

Analyses of allometric growth allow to recognize similar ontogenetic changes in *C. paleopatagonica* (*Ihering, 1903*) and *C. kokeni* (*Ihering, 1899*). In both species the ontogeny is characterized by the presence of positive allometry in the growth of posterior end, resulting in elongated adult shells. The species *Venericardia camachoi* Vigilante, 1977 and *C. patagonicus* (*Ihering, 1907*), proposed as synonyms of both previously mentioned taxa, fall into the portion of the resulting morphospace that represents juvenile morphologies, so that the obtained results corroborate these synonymies.

This particular allometric growth, resulting in elongated adult shells, is presumed in other infaunal bivalve groups (e.g., Veneridae, Trigoniidae, Carditidae and Crassatellidae). The recognition of this character has taxonomic, ecologic and evolutionary implications, being important as the starting point for further heterochronic studies in bivalves. This study includes new observations and discussion about allometric growth in infaunal bivalves, and represented a contribution for thickening the literature of cases of allometric patterns.

## ACKNOWLEDGEMENTS

The authors are in debt to curators M. Longobucco (MACN) and M. Tanuz (CPBA) who facilitated the access to collections. We thank to C. del Río (MACN) and F. Prevosti for their helpful comments in early stages of this work. We are in debt to the editor, K. De Baets (Friedrich-Alexander Universität Erlangen-Nürnberg), and the reviewers P. Milla Carmona (CPBA) and R. La Perna (Universitá di Bari), whose suggestions greatly improved this work. P. Milla Carmona suggested the lines at the end of introduction section.

### Funding
The authors were supported by CONICET (Argentina) through doctoral and post-doctoral grants. The funders had no role in study design, data collection and analysis, decision to publish, or preparation of the manuscript.

### Grant Disclosures
The following grant information was disclosed by the authors:
CONICET.

### Competing Interests
The authors declare there are no competing interests.

## Author Contributions

- Damián Eduardo Perez and María Belén Santelli conceived and designed the experiments, performed the experiments, analyzed the data, contributed reagents/materials/analysis tools, prepared figures and/or tables, authored or reviewed drafts of the paper, approved the final draft.

## Data Availability

The raw data are provided in the Supplemental Files.

## Supplemental Information

Supplemental information for this article can be found online at http://dx.doi.org/10.7717/peerj.5051#supplemental-information.

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
