# Peer review of "Allometric shell growth in infaunal burrowing bivalves: examples of the archiheterodonts Claibornicardia paleopatagonica (Ihering, 1903) and Crassatella kokeni Ihering, 1899"

_PeerJ, doi:10.7717/peerj.5051_

## Round 0.1 · original submission · Major Revisions

This is nice contribution to the literature addressing allometry patterns in fossil bivalves using an “longitudinal” approach , which is would like to see published. There are however some crucial issues which need to be addressed before publication:

Statements in the introduction: I think collecting longitudinal data (sensu Klingenberg 1997) is important and still done in too little cases. However, it is wrong to say that it is impossible in most fossil studies – it is available in many groups with accretionary growth (e.g., shelled mollusks like gastropods (Urdy et al. 2010), brachiopods (Rudwick 1968; Ackerly 1989; Tomašových, Carlson Sandra & Labarbera 2008), etc.), so it is rather that too little authors study it. It is for example increasingly used in now extinct groups like ammonoids (Korn 2012; De Baets, Klug & Monnet 2013; Korn 2017). Another issue might that in paleobiological studies sample size is comparatively small (Brown & Vavrek 2015). Although in mollusks where the entire ontogeny is often preserved – taken multiple measurement pro specimen might be an effective way to obtain more measurements at individual sizes (De Baets, Klug & Monnet 2013) and less biased as you track the same individual during its growth. How can we be sure in a cross-sectional approach in groups lacking accretionary growth like vertebrate and arthropods that the juveniles you measure are an accurate reflection of how the adults you actually measured looked like when they were juveniles in small samples.

Longitudinal versus cross-sectional studies: I feel that using the terms longitudinal versus cross-sectional studies sensu Klingenberg is appropriate here. However, I also agree with the reviewer 1 that the terms are somehow confusing when studying fossil organisms. This has also been pointed out by Korn (2017) for coiled mollusks where “longitudinal” data is usually collected by using cross-sections. In a similar note, the term longitudinal could refer to the long axis. Therefore, I would advise to always add “(sensu Klingenberg 1996)” when these terms cross-sectional or longitudinal are used in the sense of studying allometry and size through ontogeny taking multiple measurements in the same individuals.

Size classes: Sometimes it might also help to see how parameters changes through ontogeny in relationship with particular sample sizes of particular ontogenetic size or size classes. This could be assessed by plotting histograms (Monnet et al. 2010) or more appropriately violin or beanplots through ontogeny for particular ontogenetic stages or logarithmic size classes (e,g,, De Baets et al. 2013). If you get a unimodal( gaussian) distribution for each stage/class, you would have further support that they might belong to a single species and you could plot in which position juveniles described as separate species plot within it. Yes, i know these are examples from ammonoids, but these are i think nice examples of how it has been done previously. These are just one approach - there might be much more possibilities also in the bivalve literature.

Scientific reproducibility: you need to clearly describe how you performed your analyses (see specifically comments by reviewer 1 pertaining to acquirement and processing of your data; reviewer 2 with respect to your MRA). I agree with reviewer 2 that your manuscript would benefit from adding tables with the used data into the main manuscript. Most importantly, you also need to make sure your analyses are reproducible – reviewer 2 could not reproduce your results with the provided data.

Taxonomic implications: I agree with reviewer 1 that the taxonomic implications could be discussed in greater detail by for example plotting the position of synonimizable species (e.g., Monnet et al. 2010)

Language issues: I agree with reviewer 2 that there are some language issues. Please let an English native speaker read your manuscript before resubmission.

As some of these issues related to reproducibility, they might involve re-running or performing additional analyses, hence my decision of major revisions. However, I do think these changes should be easy to implement and should not take too long. I look forward to receiving your revised manuscript.

In addition to the suggestions of the reviewers, please also address the additional suggestions in the annotated pdf.

Suggested references:
Ackerly SC. 1989. Kinematics of Accretionary Shell Growth, with Examples from Brachiopods and Molluscs. Paleobiology 15:147-164.

Brown CM, and Vavrek MJ. 2015. Small sample sizes in the study of ontogenetic allometry; implications for palaeobiology. PeerJ 3:e818.

De Baets K, Klug C, and Monnet C. 2013. Intraspecific variability through ontogeny in early ammonoids. Paleobiology 39:75-94.

Korn D. 2012. Quantification of ontogenetic allometry in ammonoids. Evolution & Development 14:501-514.

Korn D. 2017. Goniatites sphaericus (Sowerby, 1814), the archetype of Palaeozoic ammonoids: a case of decreasing phenotypic variation through ontogeny. PalZ 91:337-352.

Monnet C, Bucher H, Wasmer M, and Guex J. 2010. Revision of the genus Acrochordiceras Hyatt, 1877 (Ammonoidea, Middle Triassic): morphology, biometry, biostratigraphy and intra-specific variability. Palaeontology 53:961-996.

Rudwick MJS. 1968. Some analytic methods in the study of ontogeny in fossils with accretionary skeletons. Journal of Paleontology 42:35-49.

Tomašových A, Carlson Sandra J, and Labarbera M. 2008. Ontogenetic nice shift in the brachiopod terebratalia transversia: relationship between the loss of rotation ability and allometric growth. Palaeontology 51:1471-1496.

Urdy S, Goudemand N, Bucher H, and Chirat R. 2010. Growth-dependent phenotypic variation of molluscan shells: Implications for allometric data interpretation. Journal of Experimental Zoology Part B: Molecular and Developmental Evolution 314 B:303-326.

·

Basic reporting

The ms is fairly well written, supported by literature references, well structured and with good illustrations.
Minor parts could be deleted/shortened.
The usage of "infaunal burrowing" is redundant, as explained in the notes.

Experimental design

Allometric growth is well known among bivalves, as discussed in the ms. However, the work offer a valuable contribution to shape analysis based on "longitudinal data". I don't think "longitudinal" is a good term, but the method is interesting and only scantly used in palaeontology.
Methods, from data acquisition to processing should be more clearly reported and/or explained, in particular:
1) Concerning the acquisition of growth-related shell outlines from a single valve, the only explanation is "digitized in different angles". It is very difficult to orientate valves in order to capture different growth stages (growth line). Are authors sure that outline acquisition was not biased by wrong orientation? Which method did they follow for handling and properly orienting valves?
2) Authors should give more information about the software used to digitize outlines. Further, more importantly, they should specify how many points were used, and how/where the starting point was selected. It is very difficult to find a starting point along a bivalve outline, particularly when the bivalve shape lacks any sharp/pointed traits which can offer a reliable starting point.
More unclear points about methods are remarked in the text.

Validity of the findings

Findings and implications are interesting, though not particularly new or useful, since the allometric growth is well known for bivalves.
Probably the most innovative contribution is the method ("longitudinal data") as reported above. For this reason, shape analysis methodology should be more carefully explained and illustrated.
Palaeoecological implications of allometry, though supported by data, are not a novelty.
Taxonomic implications are interesting, and could deserve to be more exhaustively discussed.

·

Basic reporting

The article is a correctly structured piece of research. The references used provide good context for the reader. Figures are ok (there are some issues with the captions though, pointed out in the attached PDF file). I would like to have a table or two reporting the results from some analyses in the main article (see Experimental design). On the negative side, I was not able to replicate the results reported in the manuscript using the supplementary data provided by the authors. However, my main problem with this article is its deficient English, which needs substantial polishing in order to make the manuscript publishable (I have made several corrections and comments in the attached PDF file aimed in that direction).

Experimental design

I think this work is good in both intention and execution, as it addresses the study of allometric growth patterns using longitudinal data (a relevant topic in paleobiology that requires accumulation of study cases like this) using an adequate quantitative toolkit (geometric morphometrics and multivariate statistics) and discussing relevant/interesting implications (taxonomic, paleobiological, evolutionary). However, there are some issues that need to be addressed. My main concerns are (see also comments in the attached PDF):

i) Some passages of the manuscript need further elaboration. In particular the geometric morphometrics paragraph at the end of the Introduction section, as well as discussion relating these study cases to the study of heterochrony at the end of the Discussion section, should be expanded.

ii) The report of multivariate regression analysis (MRA) is problematic. The authors do not mention the software they used, and the results reported are not those of an ordinary MRA (e.g., permutations and linear combination of response variables are not typical elements of an ordinary MRA). More thoroughness is needed in this regard to fully understand what the authors did. Also, I think that the information provided in Supplementary Data S4 and S5 regarding MRA should be reorganized as one or two tables and included in the main article, as these results are crucial to describe the covariation between size and shape in the studied taxa.

iii) The supplementary material provided by the authors does not allow replication of the procedures and results reported. In the case of Supplementary Data S2, some of the normalized (as I can infer from the name and number of columns) coefficient values from the last columns can not be read. In the case of Supplementary Data S3, it is not clear to me whether the table contains normalized or raw Fourier coefficients: the names of the columns suggest that these have been normalized (as coefficients start at D1), whereas the number of columns (28) and the values of the first three columns of coefficients (invariant when these have been normalized) suggest these are raw. Also, I analyzed the data from Supplementary Data S3 using the Momocs package in R, and was not able to produce the same morphospace as the one shown in Figure 3B.

Apart from this I think that, in general, the quantitative tools described in the manuscript have been adequately applied to answer a well defined and relevant question.

Validity of the findings

I think this work is a good addition to the literature addressing allometry patterns in fossil species. Concerns regarding MRA aside, the analyses used are an adequate choice for conducting a study of this nature, and interpretation of the results is correct. Findings are appropriately placed within the paleontological context of the study case, and their implications explored to a reasonable extent (with the exception of the discussion of heterochrony mentioned in Experimental design). There is nothing in the conclusions that is not justified from what the authors report. The results and conclusions seem valid to me, although the issue of reproducibility needs to be addressed to be completely sure.

Additional comments

Please check the manuscript and supplementary materials carefully and introduce the changes needed to improve the overall readability, comprehensibility and reproducibility of this work.

---

## Round 0.2 · Minor Revisions

Thank you for making these changes - your manuscript is now easier to understand and follow. The are still some issues remaining which i would like you to address.

The most crucial points are:

Reproducibility: it is not possible using the supplementary data for C. kokeni to reproduce the axes resulting from the PCA (see comments by the reviewer). As it is possible to reproduce them for the other axis, this could just be a problem with the table uploaded in the supplementary material. This is crucial as science is all about reproducibility.

Analytical methods. The reviewer raised some issues concerning your approaches. This boils down to why did you not just do a univariate regression between size and PC1 ?; More importantly the fact that the p-values might not that informative given that you are “pseudo-replicating.” In the latter case, this could be partially resolved by adding the data from the supplementary table 4 and 5 to the main text. I agree with the reviewer that these issues are not fatal, but the assumptions and implications of these at least need to be discussed and integrated in the manuscript.

Language: Some additional typos need to be corrected and strange expressions need to be rephrased (see comments below). Letting a colleague with English mother tongue read your manuscript before resubmission would be appropriate as I am not a native speaker myself.

In addition, to these points, please also address the following points:
Line 35-37: I would be easier to follow if you would make a new sentence with everything written after the comma – Some use longitudinal data …
Line 47: I would remove De Baets, Klug & Monnet, here and add “- necessary for a more holistic view.” or something along these lines.
Line 66: this sentence sounds odd and is hard to understand; please rephrase
Line 83: not sure of “thickening” is the right term here and in other place in the manuscript
Line 85: Not clear what you mean with the “paleontological debate” – remove as it repeats what you said before
Line 96: statement instead of statement
Line 116: I would add “previously” before assigned
Line 122: 32 is not that much consider they are from different localities or where this the only ones complete enough to be used? Maybe mention this if it is the latter
Line 150: what do you mean with “image-edition software”; is this the name of the software, please mention the name rather than the type
Line 164: is it common to use different amount of harmonics when comparing taxa?
Line 184: please make sure the analysis is reproducible in other software
Line 196: reporting the p-value is fine, but integrating the supplementary tables would be better in this (see comments by reviewer)
Line 218: I guess you mean “provenance” rather than “precedence”
Line 258: replace “to extreme possibilities” with “to the extremes”
Line 287: I guess you mean “observed” rather than “appreciated”
Line 306: “some cases of heterochronies” – this part comes out of the blue here; some authors might not even know what you mean with it; please elaborate
Line 325: not sure if thickening is the right word

·

Basic reporting

The revised version of the manuscript has greatly improved in readability and clarity, especially regarding the motivations and procedures adopted by the authors. Modified versions of figures represent a nice improvement as well (I particularly liked the new version of Figure 2). I still miss one or two tables reporting the results of regression analyses in the main article (I explain the reason below). I was able to repeat the main analyses the authors perform, obtaining the same results reported in the manuscript for the case of C. paleopatagonica but not for C. kokeni, something that needs to be addressed. I made some corrections in the attached PDF file, but these are all minor.

Experimental design

Having been able to properly analyze the data, I have two main concerns regarding the analytical approach adopted by the authors. The following comments are valid for the case of C. paleopatagonica, but are applicable to the case of C. kokeni as well:

1) The approach adopted by the authors is only superficially multivariate. Although three principal components are retained and analyzed by means of a multivariate regression analysis (MRA), the authors are focused exclusively on the changes along PC1, as both the figures and discussion provided reveal. I do not think this is a bad thing per se, especially considering that the shape transformation represented by PC1 is not only the main axis of variation in its own right, but also contains the allometric variation the authors are interested in. However, by including PC2 and PC3 in the analyses but not in the subsequent discussion, the authors give an unnecessarily complicated (and potentially misleading) picture of their study. I think that, given the primary interest of this work, a univariate regression between size and PC1 is both more adequate and easy to interpret.

2) By performing their MRA approach using the pooled sample of outlines, without structuring their analyses by individuals, the authors are effectively pseudo-replicating (i.e., several outlines can be extracted from the same individual, but these outlines are not statistically independent from one another), something that undermines the validity of the obtained p-value. Performing regressions in this way can still be valuable if a more nuanced, descriptive interpretation is attempted (i.e., looking for general trends in the values of slopes and intercepts rather than the concrete p-value for these coefficients), although the reason behind this decision should be clearly stated. This is why I think the tables from Supplemental Data S4 and S5 are far more useful than the reported p-value, and thus should be included in the main article (rather than be relegated to the supplementary materials).

I do not think these flaws are fatal. I would recommend the authors to perform a univariate regression (one for each species) using PC1 as response variable and focus in the interpretation of slopes (p-values can still be reported, although as I already mentioned they need to be interpreted with caution). I am including this analysis in the R script I wrote to replicate the procedures. Alternatively, in the case the authors choose to retain the MRA approach, I would like to see a little more attention given to PC2 and PC3, both in the discussion of the significance of these shape transformations and in the figures (i.e., provide visual representations of the shape transformations implied by PC2 and PC3).

In addition, I was not able to replicate the results reported for C. kokeni using the supplementary data provided by the authors (Supplemental Data S3). In short, the axes resulting from the PCA of the normalized coefficients stored in this table are different from the ones depicted in the Figures (Figs 3B–C) (I performed the analyses in the R environment using the basic functions; the procedures can be readily repeated using the R script included in the general comments for the authors. This same script was used to correctly replicate the results reported in the manuscript for C. paleopatagonica). I assume the problem is the uploaded data; if so, replacing the file should be enough. However, if this discrepancy stems from an incorrect analysis the problem could be more serious, as this could compromise the conclusions of the study.

Validity of the findings

The results and conclusions seem well founded, and the implications are discussed to a sufficient extent. As noted above, Supplemental Data S3 must be replaced by the file containing the actual coefficients used by the authors to perform their analyses.

Additional comments

## R script

## import the data (as csv file and skipping the first two lines)
data<-read.csv("/home/pablo/Insync/peerj-25256/supplemental/peerj-25256-Supplemental_Data_S2.csv", skip=2)
str(data)
head(data)
dim(data)

## store the coefficients, size and classification of the sample in different objects
coe<-data[,5:ncol(data)]
size<-data[,2]
class<-data[,4]

## principal component analysis on the matrix of coefficients
pca<-prcomp(coe)

## plot morphospace
plot(pca$x[,1:2], pch=16, col=c("red", "blue","black")[class])

## multivariate regression of PC1, PC2 and PC3 on size
mra<-manova(pca$x[,1:3]~size)
summary(mra)

## univariate regression of PC1 on size
ur<-lm(pca$x[,1]~size)
summary(ur) ## results

## plot the regression
plot(size, pca$x[,1], pch=16, xlab="Size", ylab="PC1")
abline(ur, col="red")

---

## Round 0.3 · Minor Revisions

Thank you for addressing all our suggestions including making sure the analyses can be reproduced in both Past and R for C. kokeni from the data listed in the supplementary data. Your manuscript is as good as accepted.

I just had some minor additional suggestions on the revised manuscript which I would like to care off before publication. As you appropriately list that you analyzed your data both with Past and R, it would be in order to state more clearly which analyses have been performed with Past and which ones in R (just PCAs ?). Furthermore, if you used R, it would be appropriate to list which packages (if any) have been used for analysis . Ideally, one would also make the R script available for the sake of reproducibility.

Line 174: “y” should be replaced by “and”.

Line 335: “W De Baets” should be replaced by “K. De Baets”

---

## Round 0.4 · accepted · Accept

Thank you for implementing these final suggestions, particularly stating more clearly which software was used for which analysis and adding the R-script in the supplementary material.

#